# Learning to Learn via Self-Critique

**Antreas Antoniou**
University of Edinburgh
{a.antoniou}@sms.ed.ac.uk

**Amos Storkey**
University of Edinburgh
{a.storkey}@ed.ac.uk

## Abstract

In few-shot learning, a machine learning system learns from a small set of labelled examples relating to a specific task, such that it can generalize to new examples of the same task. Given the limited availability of labelled examples in such tasks, we wish to make use of all the information we can. Usually a model learns task-specific information from a small training-set (*support-set*) to predict on an unlabelled validation set (*target-set*). The target-set contains additional task-specific information which is not utilized by existing few-shot learning methods. Making use of the target-set examples via transductive learning requires approaches beyond the current methods; at inference time, the target-set contains only unlabelled input data-points, and so discriminative learning cannot be used. In this paper, we propose a framework called *Self-Critique and Adapt* or SCA, which learns to learn an label-free loss function, parameterized as a neural network. A base-model learns on a support-set using existing methods (e.g. stochastic gradient descent combined with the cross-entropy loss), and then is updated for the incoming target-task using the learnt loss function. The label-free loss function is learned such that the target-set-updated model achieves higher generalization performance. Experiments demonstrate that SCA offers substantially reduced error-rates compared to baselines which only adapt on the support-set, and results in state of the art benchmark performance on Mini-ImageNet and Caltech-UCSD Birds 200.

## 1 Introduction

Humans can learn from a few data-points and generalize very well, but also have the ability to adapt in real-time to information from an incoming task. Given two training images for a two class problem, one with a cat on a white sofa, and one with a dog next to a car, one can come up with two hypothesis as to what each class describes in each case. A test image that presents a dog with a cat would be ambiguous. However, having other unlabelled test images with cats in other contexts and cars in other contexts enables these cases to be disambiguated, and it is possible to learn to focus on features that help this separation. We wish to incorporate this ability to adapt into a meta-learning context.

Few-shot learning is a learning paradigm where only a handful of samples are available to learn from. It is a setting where deep learning methods previously demonstrated weak generalization performance. In recent years, by framing the problem of few-shot learning as a meta-learning problem (Vinyals et al., 2016), we have observed an advent of meta-learning methods that have demonstrated unprecedented performance on a number of few-shot learning benchmarks (Snell et al., 2017; Finn et al., 2017; Rusu et al., 2018).

Most few-shot meta-learning methods have focused on either learning static (Finn et al., 2017; Antoniou et al., 2019; Li and Malik, 2016) or dynamic parameter initializations (Rusu et al., 2018), learning rate schedulers (Antoniou et al., 2019), embedding functions Vinyals et al. (2016); Snell et al. (2017), optimizers Ravi and Larochelle (2016) and other internal components. However all of

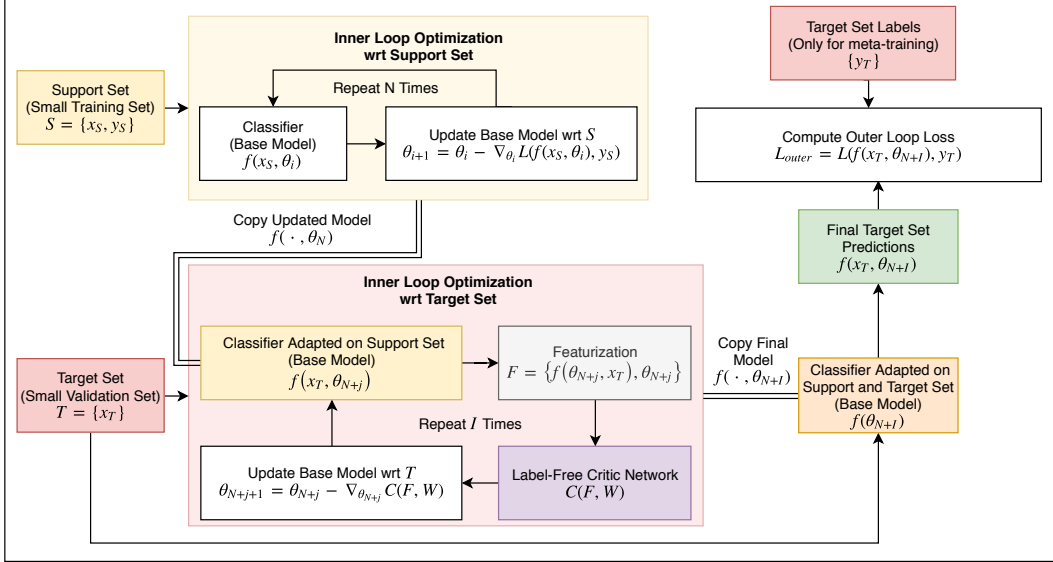

Figure 1: Proposed Method. Starting from the top-left, task-specific knowledge from the support-set is used to train the base model, updating $\theta_0$ to $\theta_N$. At this point, standard meta-learning methods return predictions from this learnt model to complete their inference. Instead, we use an unsupervised loss from a critic network $C$ applied to the unlabelled target set to make further updates. To do this we collect a set of features $F$ that summarise the model applied to the target set $T$; these features are sent to the critic $C$, a neural network with parameters $W$. Using the loss from this critic network, and the model with parameters $\theta_N$, we make further updates to get $\theta_{N+I}$. We use the predictions from this model as our predictions for the target set. During training, an *outer-loop* loss comparing target-set predictions to target set labels is used to update the initial parameters $\theta_0$ and the critic model parameters $W$.

these methods explore learning from the labelled support-set, whereas learning (at inference time) from the unlabelled target-set has remained unexplored.[1]

In this paper, we propose a mechanism we call *Self-Critique and Adapt* or SCA that enables meta-learning-based few-shot systems to learn not only from the support-set input-output pairs, but also from the target-set inputs, by learning a label-free loss function, parameterized as a neural network. Doing so grants our models the ability to learn from the target-set input data-points, simply by computing a loss, conditioned on base-model predictions of the target-set. The label-free loss can be used to compute gradients with respect to the model, and the gradients can then be used to update the base-model at inference time, to improve generalization performance. Furthermore, the proposed method can be added on top of any modern meta-learning method, including both methods that utilize gradient updates on the support set, such as MAML (Finn et al., 2017), as well as ones that do not use gradient-based updates on the support set, such as Matching Networks (Vinyals et al., 2016).

Self-Critique and Adapt is a *transductive learning* approach (Vapnik, 2006). Transductive learning uses training data and test input data-points to learn a model that is **specifically** tuned to produce predictions for the given test-set. Transductive learning benefits from *unsupervised information* from the test example points, and *specification* by knowing where we need to focus model capability. In stark contrast, *inductive learning*, can be defined as a learning paradigm where given training input-output pairs, a model is learned consisting of **general** rules, that can then be used on any test-set without refinement to produce predictions. Given that, in a meta-learning context, additional learning needs to be done for each new setting anyway, and given the importance of making the most of every piece of information, transductive learning is a natural learning paradigm for the few-shot learning setting.

We evaluate the proposed method on the established few-shot learning benchmarks of Mini-ImageNet (Ravi and Larochelle, 2016) and Caltech-UCSD Birds 200 (CUB) (Chen et al., 2019). The evaluation results indicate that our method substantially boosts the performance of two separate instances of the MAML++ (Antoniou et al., 2019) framework, setting a new state-of-the-art performance for all tasks in both benchmarks.

This paper's contributions are:

1. An approach that gives state-of-the-art performance in the Mini-Imagenet and Caltech-UCSD Birds 200 (CUB) benchmark tasks by using both support and target set information through a transductive approach.

2. The ability to learn to learn a flexible parameterized loss function appropriate for a supervised problem but defined on unlabelled data; this loss function can be used to enhance training on semi-supervised data.

3. A set of ablation studies on different conditioning features for the critic network, revealing which features are most useful to the few-shot learning benchmarks.

## 2 Related Work

The *set-to-set* few-shot learning setting (Vinyals et al., 2016) has been vital in framing few-shot learning as a meta-learning problem. In set-to-set few-shot learning, we have a number of tasks, which a model seeks to learn. Each task is composed of two small sets. A small training-set or *support-set* used for acquiring task-specific knowledge, and a small validation-set or *target-set*, which is used to evaluate the model once it has acquired knowledge from the support-set. The tasks are generated dynamically from a larger dataset of input-output pairs. The dataset is split into 3 subsets beforehand, the *meta-training*, *meta-validation* and the *meta-test* sets, used for training, validation and testing respectively.

Once meta-learning was shown to be very effective in learning to few-shot learn, a multitude of methods showcasing unprecedented performance in few-shot learning surfaced. Matching Networks (Vinyals et al., 2016), and their extension Prototypical Networks (Snell et al., 2017) were some of the first methods to showcase strong few-shot performance, using learnable embedding functions parameterized as neural networks in combination with distance metrics, such as cosine and euclidean distance. Unsupervised and supervised set-based embedding methods were also developed for the few-shot setting (Edwards and Storkey, 2017).

Further advancements were made by gradient-based meta-learning methods that explicitly optimized themselves for fast adaptation with a few data-points. The first of such methods, Meta-Learner LSTM (Ravi and Larochelle, 2016) attempted to learn a parameter-initialization and an optimizer, parameterized as neural networks for fast adaptation. Subsequently, additional improvements came from *Model Agnostic Meta-Learning* (MAML) (Finn et al., 2017) and its improved versions *Meta-SGD* (Li et al., 2017) and *MAML++* (Antoniou et al., 2019), where the authors proposed learning a parameter-initialization for a base-model that is adapted with standard SGD for a number of steps towards a task.

Furthermore, Relational Networks (Santoro et al., 2017), that were originally invented for relational reasoning, demonstrated very strong performance in few-shot learning tasks (Santurkar et al., 2018). The effectiveness of relational networks in a large variety of settings made them a module often found in meta-learning systems.

In the Reinforcement Learning domain, meta-learning has been used to learn an unsupervised loss function for sample-efficient adaptation in (Yu, 2018). Their work differs from ours, in that our model adapts a meta-learned initialization by both labelled and unlabelled examples; we do use a learnt unsupervised model, but the combination is critical. Their work is purely unsupervised in the inner loop, using RL as the outer loop, whereas ours consists of both supervised and unsupervised inner loops and a supervised outer loop. Furthermore, in (Houthooft et al., 2018), the authors propose a method that learns a state and reward conditional loss function to train a policy network, using RL inner phase and an evolutionary algorithm outer loop. Our work, instead targets a different problem, that is few-shot learning, using a supervised outer loop, and a transductive inner loop (composed by supervised and unsupervised inner loop phases). Furthermore, in (Sung et al., 2017)

the authors propose learning a supervised loss function for few-shot learning, as well as a state and reward conditional loss function for training RL agents.

Shortly after, substantial progress was made using a hybrid method utilizing embeddings, gradient-based methods and dynamic parameter generation called Latent Embedding Optimization Rusu et al. (2018).

Semi-supervised learning via learning label-free functions were also attempted in Rinu Boney (2018). Transductive learning for the few-shot learning setting has previously been attempted by learning to propagate labels (Liu et al., 2018). Their work differs from ours in that we learn to transduce using a learned unsupervised loss function whereas they instead learn to generate labels for the test set.

---

**Algorithm 1** SCA Algorithm combined with MAML

---

1: **Required** : Base model function $\mathbf{f}$ and initialisation parameters $\boldsymbol{\theta}$, critic network function $\mathbf{C}$ and parameters $\mathbf{W}$, a batch of tasks $\{\mathbf{S^B} = \{x_S^B, y_S^B\}, \mathbf{T^B} = \{x_T^B, y_T^B\}\}$ (where $\mathbf{B}$ is the number of tasks in our batch) and learning rates $\alpha, \beta, \gamma$

2: $L_{outer} = 0$

3: **for** b in range(B) **do**

4: $\quad \theta_0 = \theta$ $\qquad\qquad\qquad\qquad\qquad\qquad$ ▷ Reset $\theta_0$ to the learned initialization parameters

5: $\quad$ **for** i in range(N) **do** $\qquad$ ▷ N indicates total number of inner loop steps wrt support set

6:
$$\theta_{i+1} = \theta_i - \alpha \nabla_{\theta_i} L(f(x_S^b, \theta_i), y_S^b) \tag{1}$$
$\qquad\qquad\qquad\qquad\qquad\qquad\qquad\qquad$ ▷ Inner loop optimization wrt support set

7: $\quad$ **for** j in range(I) **do** $\qquad\qquad$ ▷ I indicates total number of inner loop steps wrt target set

8:
$$F = \{f(x_T^b, \theta_{N+j}), \theta_N + j, g(x_S, x_n)\} \tag{2}$$
$\qquad\qquad\qquad\qquad\qquad\qquad\qquad\qquad$ ▷ Critic feature-set collection

9:
$$\theta_{N+j+1} = \theta_{N+j} - \gamma \nabla_{\theta_{N+j}} C(F, W) \tag{3}$$
$\qquad\qquad\qquad\qquad\qquad\qquad\qquad\qquad$ ▷ Inner loop optimization wrt target set

10: $\quad L_{outer} = L_{outer} + L(f(x_T^b, \theta_{N+I}), y_T^b)$

11:
$$\theta = \theta - \beta \nabla_\theta L_{outer} \tag{4}$$
$\qquad\qquad\qquad\qquad\qquad\qquad\qquad\qquad$ ▷ Joint outer loop optimization of $\theta$

12:
$$W = W - \beta \nabla_W L_{outer} \tag{5}$$
$\qquad\qquad\qquad\qquad\qquad\qquad\qquad\qquad$ ▷ Joint outer loop optimization of $W$

---

## 3 Self-Critique and Adapt

For a model to learn and adapt in a setting where only input data-points are available (e.g. on given task's few-shot target-set), one needs a label-free loss function. For example, many unsupervised learning approaches try to maximize the generative probability, and hence use a negative log-likelihood (or bound thereof) as a loss function. In general though, much of the generative model will be task-irrelevant. In the context of a particular set of tasks there are likely to be more appropriate, specialised choices for a loss function.

Manually inventing such a loss function is challenging, often only yielding loss functions that might work in one setting but not in another. Understanding the full implications of a choice of loss-function is not easy. Instead, we propose a Self-Critique and Adapt approach which *meta-learns* a loss function for a particular set of tasks. It does this by framing the problem using the set-to-set few-shot learning framework and using end-to-end differentiable gradient-based meta-learning as our learning framework.

SCA is model-agnostic, and can be applied on top of **any** end-to-end differentiable, gradient-based, meta-learning method that uses the inner-loop optimization process to acquire task-specific information. Many such approaches (Ravi and Larochelle, 2016; Finn et al., 2017; Li et al., 2017; Antoniou et al., 2019; Finn et al., 2018; Qiao et al., 2018; Rusu et al., 2018; Grant et al., 2018) are currently competing for state-of-the-art in the few-shot learning landscape.

Self-Critique and Adapt, summarised in Figure 1, takes a base-model, updates it with respect to the support-set with an existing gradient-based meta-learning method (e.g. MAML (Finn et al., 2017), MAML++ (Antoniou et al., 2019) or LEO (Rusu et al., 2018)), and then infers predictions for the target-set. Once the predictions have been inferred, they are concatenated along with other base-model related information (e.g. model parameters, a task embedding etc.), and are then passed to a learnable *critic loss network*, the output of which should be interpreted as a loss value for the given inputs. This critic network computes and returns a loss with respect to the target-set. The base-model is then updated with any stochastic gradient optimization method (such as SGD) with respect to this critic loss; updates can be done a number of times if necessary. This *inner-loop* optimization produces a predictive model specific to the support/target set information.

The inner loop process is used directly at inference time for the task at hand. However, as in other meta-learning settings, we optimize the inner loop using a collection of training tasks (these are different from the test tasks as explained in Section 7). The quality of the inner-loop learned predictive model is assessed using ground truth labels from the training tasks. The *outer loop* then optimizes the initial parameters and the critic loss to maximize the quality of the inner loop predictions. As with other meta-learning methods, the differentiability of the whole inner loop ensures this outer-loop can be learnt with gradient based methods.

In this paper we use MAML++ as the base method. We denote the model, which is parameterized as a neural network, by $f(\cdot, \theta)$, with parameters $\theta$, and the critic loss $C(\cdot, W)$, also a neural network with parameters $W$. We want to learn good parameters $\theta$ and $W$, such that when the model $f$ is optimized a number of steps $N$ with respect to the loss $L$ on the support-set $S_b = \{x_S, y_S\}$, and then additionally another $I$ steps towards the target-set $T_b = \{x_T\}$, using the critic loss $C$, it can achieve good generalization performance on the target-set. Here, $b$ is the index of a particular task in a batch of tasks. The full algorithm is described in Algorithm 1.

Equation 2 in Algorithm 1 defines a potential set of conditioning features $F$ that summarise the base-model and its behaviour. These features are what the unsupervised critic loss $C$ can use to tune the target set updates. Amongst these possible features, $f(\theta_N, x_T)$ are the predictions of the base-model $f$, using parameters $\theta_N$ (that is parameters updated for N steps on the support-set loss) and $g(x_S, x_n)$ is a task embedding, parameterized as a neural network, which is conditional on the support and target input data-points.

## 4 Example: A Prediction-Conditional Critic Network

**Model Inference and Training:** As explained in Sections 3 and 5 our critic network can be given access to a wide variety of conditioning information. Here, we describe a specific variant, which uses the predictions of the base-model on the target-set as the critic network's conditioning information; this simple form enables visualisation of what the critic loss returns.

In this example, SCA is applied directly to MAML++. Given a support set, $S$ and target set $T$ for a task, and initial model $f(\cdot, \theta_0)$, five SGD steps are taken to obtain the updated model $f(\cdot, \theta_5)$ using the support set's loss using Equation 1. The updated model is then applied to the target set to produce target-set predictions $f(x_T, \theta_5)$; these predictions are passed to the critic network, which returns a loss-value for each element of the target set. Aggregated, these loss-values form the critic loss. The base model $f(\cdot, \theta_5)$ is then updated for 1 step using the gradients with respect to the critic loss using Equation 3 to obtain the final model $f(\cdot, \theta_6)$. We use gradients of a whole batch of target set images to update the base model since this produces better performance. The results in the paper represent experiments where we used target sets of size 75 to learn the loss function, however one could instead randomly choose the batch size of the target set to train a loss function that generalizes on a wide range of batch sizes. At this stage our final model has learned from both the support and target sets and is ready to produce the final predictions of the target set. These are produced, and, at training time, evaluated against the ground truth. This final error is now the signal for updating the whole process: the choice of initial $\theta_0$ and the critic loss update. The gradients for both the base and critic model parameters can be computed using backpropagation as every module of the system is differentiable.

**Interrogating the Learned Critic Loss:** We investigate how the predictions of the base-model change after updates with respect to the critic loss. We generate target-set predictions using the base-model $f(\cdot, \theta_5)$ and pass those predictions to the learnt critic network to obtain a loss value for

each individual prediction. Five steps of SGD are performed with respect to the the individual critic loss. The results are shown in Figure 2.

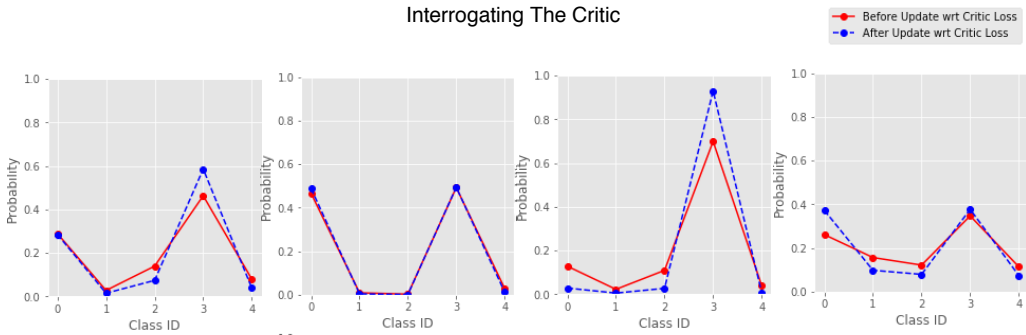

Figure 2: The target-set predictions of the base-model before (red lines) and after (green lines) it has been updated wrt the learned critic loss. Starting from the left, the first chart showcases an instance where the probabilities of two classes dominate the rest, with one class having approximately 20% higher probability than the other. The critic network proposes that the dominant class with the lower probability be left as is, whilst the higher probability class be increased. In the second chart, we present an example where two dominant classes have equal probabilities, and are thus considered optimal and left unchanged. In the third case, we present an example where a single class dominates all others. The critic network proposes that the probability of that class is pushed to almost the maximum possible value. Finally, in the fourth case we present a very interesting case where the critic, faced with two dominant classes, which had approximately 10% difference in their probability, proposed that the two classes have their probabilities matched to improve the resulting generalization performance. Note how some top classes change, e.g. in the rightmost chart. The effect of these changes is not in the class-labels alone but also allows better model initialization: training a critic on a pretrained initialization does not provide as much benefit.

## 5 Choosing Conditioning Information for the Critic Network

We find that the performance of the critic network $C$, is unsurprisingly dependent on the quality of critic features it is given access to. In this section, we will outline a set of critic features we used in experiments, from simplest to most complicated; we also discuss the intuition behind their selection. To evaluate the usefulness of each set of critic features we use these in ablation studies in Section 7 and summarize the results in Tables 1 and 2.

**Base-Model Predictions:** The predictions on the base model indicate the certainty about any input; this can be a powerful driver for an unsupervised loss. As in Section 4, given the model $f$, the support-set updated weights $\theta_N$ and some target-set data-points $x_T$, we can generate predictions $f(x_T, \theta_N)$. These predictions can be passed to our critic model $C$ to compute the loss function.

**Base-model Parameters:** Another key source of information about our base-model is the model parameters. In this context, the inner loop optimized parameters $\theta_N$ are passed to our loss network. For example this might enables it to learn *penalty* terms, similar to manually invented penalties such as *L1* and *L2*.

**Task Embeddings:** Giving the critic model direct access to task information can enable model assessment in the context of a particular task; we find this further improves performance as empirically observed in Tables 1 and 2. To generate a task embedding, we learn an embedding function $g$, parameterized as a neural network such as a DenseNet.[2] The embedding function $g$ receives the support-set images and returns a batch of embedding vectors. Taking the mean of the vectors can serve as a strong task embedding. However, we found that using a relational network to relate all embeddings with each other and then taking the sum of those embeddings produced superior results. Similar embedding functions have previously been used in (Rusu et al., 2018).

## 6 Baselines

The proposed model is, by virtue of its design, applicable to any few-shot meta-learning system that follows the set-to-set Vinyals et al. (2016) few-shot learning setting. Thus, to investigate its performance, we require baseline meta-learning systems which inner-loop optimize their parameters only on the support-set. We chose the MAML++ system, for it's simplicity and strong generalization performance.

To thoroughly evaluate our method, we experimented using two separate instances of the MAML++ framework, each differing only in the neural network architecture serving as its backbone.

**Low-End MAML++:** The backbone of our low-end baseline model follows the architecture proposed in Antoniou et al. (2019), which consists of a cascade of 4 convolutional layers, each followed by a batch normalization layer and a ReLU activation function. We optimize the entirety of the backbone during the inner loop optimization process. The low-end baseline is chosen to be identical to an existing model (MAML++), such that we could: 1. Confirm that our implementation replicates the results of the original method (i.e. makes sure that our framework does not over/under perform relatively to an existing method, and thus reduces the probability that any improvements in our results are there due to a bug in the data provider or learning framework) and 2. Investigate how our proposed method performs when added adhoc to an existing, non-tuned, architecture, therefore showcasing performance unbiased wrt architecture.

**High-End MAML++:** Methods that provide significant performance improvements on low-capacity models, often fail to provide the same level of improvement on high-capacity models and vice versa. Furthermore, meta-learning methods, have been demonstrated to be very sensitive to neural network architecture changes. Thus, to evaluate both the consistency and sensitivity of our method, we designed a high (generalization) performance MAML++ backbone. It uses a shallow, yet wide DenseNet architecture, with growth-rate 64, utilizing 2 dense-stages each consisting of two bottleneck blocks as described in Huang et al. (2017) and one transition layer. Before each bottleneck block, we apply squeeze-excite style convolutional attention as described in Hu et al. (2018), to further regularize our model. To improve the training speed and generalization performance, we restrict the network components optimized during the inner loop optimization process. In more detail, we choose to share a static copy of majority of the network components at each step, and only optimize the penultimate convolutional layer (including the squeeze excite attentional block preceding it) as well as the final linear layer. An efficient way of sharing components across steps, is to treat them as a *feature* embedding, whose features are passed to the components that will be inner loop optimized. Recent work (Rusu et al., 2018; Qiao et al., 2018) followed a similar approach. Motivations behind these design choices can be found in the supplementary materials section 1. 10.

**Critic Network Architecture:** The critic network consists of two majour components. First, a selection of conditioning features as described in Section 5, which are then reshaped into a batch of one-dimensional features vectors and concatenated on the feature dimension. Second, an *Information Integration* network, which consists of a sequence of five one-dimensional dilated convolutions with kernel size 2, and 8 kernels per convolutional layer. Further, we employ an exponentially increasing dilation policy where a given convolutional layer has dilation $d = 2^i$ where $i$ is the index of the convolutional layer in the network, starting from $0$ for the first layer and increasing up to $4$ for the fifth layer. We use Dense-Net style connectivity for our convolutional layers, more specifically, the inputs for a given layer consist of a concatenation of the output features of all preceding convolutional layers. Finally we apply a sequence of two fully connected layers using ReLU non-linearities before the final linear layer which outputs the loss value.

## 7 Experiments

To evaluate the proposed methods we first establish baselines on both the low-end and high-end variants of MAML++ on the Mini-ImageNet and Caltech-UCSD Birds 200 (CUB) 5-way 1/5-shot tasks. Then, we add the proposed critic network, which enables the adaptation of the base-model on a given target-set. To investigate how the selection of conditional information we provide to the critic affect its performance we conduct ablation studies. Due to the large number of combinations possible, we adopt a hierarchical experimentation style, where we first ran experiments with a single source of conditional information, and then tried combinations of the best performing methods.

Finally, we ran experiments where we combined every proposed conditioning method to the critic, as well as experiments where we combined every conditional method except one. Tables 1 and 2 show ablation and comparative studies on Mini-Imagenet and CUB respectively.

## 7.1 Results

The results of our experiments showcased that our proposed method was able to substantially improve the performance of both our baselines across all Mini-ImageNet and CUB tasks.

More specifically, the original MAML++ architecture (i.e. Low-End MAML++), yielded superior performance when we provided additional conditional information such as the task embedding and the model parameters. The only source of information that actually decreased performance were the base-model's parameters themselves. Furthermore, it appears that a very straight-forward strategy that worked the best, was to simply combine all proposed conditioning sources and pass them to the critic network.

However, in the case of the High-End MAML++ architecture, performance improvements were substantial when using the predictions and the task embedding as the conditioning information. Contrary to the results on MAML++, providing the critic with the matching network and relational comparator features (in addition to the prediction and task embedding features) did not produce additional performance gains.

| Model | Test Accuracy | | | |
| | Mini-Imagenet | | CUB | |
| | 1-shot | 5-shot | 1-shot | 5-shot |
|---|---|---|---|---|
| MAML++ (Low-End) | $52.15 \pm 0.26\%$ | $68.32 \pm 0.44\%$ | $62.19 \pm 0.53\%$ | $76.08 \pm 0.51\%$ |
| MAML++ (Low-End) with SCA(preds) | $52.52 \pm 1.13\%$ | $70.84 \pm 0.34\%$ | $66.13 \pm 0.97\%$ | $77.62 \pm 0.77\%$ |
| MAML++ (Low-End) with SCA(preds, params) | $52.68 \pm 0.93\%$ | $69.83 \pm 1.18\%$ | - | - |
| MAML++ (Low-End) with SCA(preds, task-embedding) | $\mathbf{54.84 \pm 1.24}\%$ | $70.95 \pm 0.17\%$ | $65.56 \pm 0.48\%$ | $77.69 \pm 0.47\%$ |
| MAML++ (Low-End) with SCA(preds, task-embedding, params) | $54.24 \pm 0.99\%$ | $\mathbf{71.85 \pm 0.53}\%$ | - | - |
| MAML++ (High-End) | $58.37 \pm 0.27\%$ | $75.50 \pm 0.19\%$ | $67.48 \pm 1.44\%$ | $83.80 \pm 0.35\%$ |
| MAML++ (High-End) with SCA (preds) | $\mathbf{62.86 \pm 0.70}\%$ | $77.07 \pm 0.19\%$ | $70.33 \pm 0.78\%$ | $85.47 \pm 0.40\%$ |
| MAML++ (High-End) with SCA (preds, task-embedding) | $62.29 \pm 0.38\%$ | $\mathbf{77.64 \pm 0.40}\%$ | $\mathbf{70.46 \pm 1.18}\%$ | $\mathbf{85.63 \pm 0.66}\%$ |

Table 1: SCA Ablation Studies on Mini-ImageNet and CUB: All variants utilizing the proposed SCA method perform substantially better than the non-SCA baseline variant. Interestingly, the best type of critic conditioning features varies depending on the backbone architecture. Based on our experiments, the best critic conditioning features for the Low-End MAML++ is the combination of predictions, task-embedding and network parameters, whereas on High-End MAML++, using just the target-set predictions appears to be enough to obtain the highest performance observed in our experiments.

| Model | Test Accuracy | | | |
| | Mini-ImageNet | | CUB | |
| | 1-shot | 5-shot | 1-shot | 5-shot |
|---|---|---|---|---|
| Matching networks (Vinyals et al., 2016) | $43.56 \pm 0.84\%$ | $55.31 \pm 0.73\%$ | $61.16 \pm 0.89\%$ | $72.86 \pm 0.70\%$ |
| Meta-learner LSTM (Ravi and Larochelle, 2016) | $43.44 \pm 0.77\%$ | $60.60 \pm 0.71\%$ | - | - |
| MAML (Finn et al., 2017) | $48.70 \pm 1.84\%$ | $63.11 \pm 0.92\%$ | $55.92 \pm 0.95\%$ | $72.09 \pm 0.76\%$ |
| LLAMA (Grant et al., 2018) | $49.40 \pm 1.83\%$ | - | - | - |
| REPTILE (Nichol et al., 2018) | $49.97 \pm 0.32\%$ | $65.99 \pm 0.58\%$ | - | - |
| PLATIPUS (Finn et al., 2018) | $50.13 \pm 1.86\%$ | - | - | - |
| Meta-SGD (our features) | $54.24 \pm 0.03\%$ | $70.86 \pm 0.04\%$ | - | - |
| SNAIL (Mishra et al., 2017) | $55.71 \pm 0.99\%$ | $68.88 \pm 0.92\%$ | - | - |
| (Gidaris and Komodakis, 2018) | $56.20 \pm 0.86\%$ | $73.00 \pm 0.64\%$ | - | - |
| (Munkhdalai and Yu, 2017) | $57.10 \pm 0.70\%$ | $70.04 \pm 0.63\%$ | - | - |
| TADAM (Oreshkin et al., 2018) | $58.50 \pm 0.30\%$ | $76.70 \pm 0.30\%$ | - | - |
| (Qiao et al., 2018) | $59.60 \pm 0.41\%$ | $73.74 \pm 0.19\%$ | - | - |
| LEO (Rusu et al., 2018) | $61.76 \pm 0.08\%$ | $77.59 \pm 0.12\%$ | - | - |
| Baseline (Chen et al., 2019) | - | - | $47.12 \pm 0.74\%$ | $64.16 \pm 0.71\%$ |
| Baseline ++ (Chen et al., 2019) | - | - | $60.53 \pm 0.83\%$ | $79.34 \pm 0.61\%$ |
| MAML (Local Replication) | $48.25 \pm 0.62\%$ | $64.39 \pm 0.31\%$ | - | - |
| MAML++ (Low-End) | $52.15 \pm 0.26\%$ | $68.32 \pm 0.44\%$ | $62.19 \pm 0.53\%$ | $76.08 \pm 0.51\%$ |
| MAML++ (Low-End) + SCA | $54.84 \pm 0.99\%$ | $71.85 \pm 0.53\%$ | $66.13 \pm 0.97\%$ | $77.62 \pm 0.77\%$ |
| MAML++ (High-End) | $58.37 \pm 0.27\%$ | $75.50 \pm 0.19\%$ | $67.48 \pm 1.44\%$ | $83.80 \pm 0.35\%$ |
| **MAML++ (High-End) + SCA** | $\mathbf{62.86 \pm 0.79}\%$ | $\mathbf{77.64 \pm 0.40}\%$ | $\mathbf{70.46 \pm 1.18}\%$ | $\mathbf{85.63 \pm 0.66}\%$ |

Table 2: Comparative Results on Mini-ImageNet and CUB: The proposed method appears to improve the baseline model by over 4 percentage points, allowing it to set a new state-of-the-art result on both the 1/5-way Mini-ImageNet tasks.

## 8 Conclusion

In this paper we propose adapting few-shot models not only on a given support-set, but also on a given target-set by learning a label-free critic model. In our experiments we found that such a critic network can, in fact, improve the performance of two instances of the well established gradient-based meta-learning method MAML. We found that some of the most useful conditional information for the critic model were the base-model's predictions, a relational task embedding and a relational support-target-set network. The performance achieved by the High-End MAML++ with SCA is the current best across all SOTA results. The fact that a model can learn to update itself with respect to an incoming batch of data-points is an intriguing one. Perhaps deep learning models with the ability to adapt themselves in light of new data, might provide a future avenue to flexible, self-updating systems that can utilize incoming data-points to improve their own performance at a given task.

## 9 Acknowledgements

We would like to thank our colleagues Elliot Crowley, Paul Micaelli, James Owers and Joseph Mellor for reviewing this work and providing useful suggestions/comments. Furthermore, we'd like to thank Harri Edwards for the useful discussions at the beginning of this project and the review and comments he provided. This work was supported in part by the EPSRC Centre for Doctoral Training in Data Science, funded by the UK Engineering and Physical Sciences Research Council (grant EP/L016427/1) and the University of Edinburgh as well as a Huawei DDMPLab Innovation Research Grant.

## Footnotes

[1]Existing techniques like MAML Finn et al. (2017) utilize target-set information by computing means and standard deviations for the batch normalization layers within their base models. However, we don't consider that as explicit learning, but instead, as a minimal adaptation routine.

[2]Here we use a DenseNet, with growth rate 8, and 8 blocks per stage, for a total of 49 layers. The DenseNet is reqularised using dropout after each block (with drop probability of 0.5) and weight decay (with decay rate 2e-05).

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

## 10   Appendix: High-End Backbone details:

The motivations behind each of the design choices can be found below.

1. Using DenseNet as the backbone, which decreases probability of gradient degradation problems and by allowing feature-reuse across all blocksm improves parameter/training efficiency. MAML is highly vulnerable to gradient degradation issues, and thus ensuring that our backbone decreases probability of such issues is of vital importance.

2. Using a shallow, yet wide backbone: Previous works Qiao et al. (2018); Rusu et al. (2018) have demonstrated that using features from the 20th layer of a pretrained ResNet produces superior generalization performance. The authors made the case that using features from shallower parts of the network decreases the probability that the features are too class-specific, and thus allow for better generalization on previously unseen classes. In both Qiao et al. (2018); Rusu et al. (2018) the authors did not train their meta-learning system end-to-end, and instead trained the feature backbone and the meta-learning components separately.

However, in preliminary experiments we found that ResNet and DenseNet backbones tend to overfit very heavily, and in pursuit of a high-generalization end-to-end trainable meta-learning system, we experimented with explicit reduction of the effective input region of the layers in a backbone. Doing so, ensures that features learned will be local. We found that keeping the effective input region of the deepest layer to approximately 15x15/20x20 produced the best results for both Mini-ImageNet and CUB. Furthermore, we found that widening the network produced additional generalization improvements. We theorize that this is because of a higher probability for a randomly initialized feature to lie in just the right subspace to produce a highly generalizable feature once optimized.

3. Using bottleneck blocks, preceded by squeeze-excite-styleHu et al. (2018) convolutional attention: We empirically found that this improves generalization performance.

4. Inner-Loop optimize only the last squeeze excite linear layer, as well as the last convolutional layer and the final linear layer, whilst sharing the rest of the backbone across the inner loop steps. This design choice was hinted by the learned per-layer, per-step learning rates learned by MAML++ on the low-end baseline. More specifically, the learned learning rates where close to zero, for all layers, in all steps, except the last convolutional and last linear layers. Thus, we attempted to train a MAML++ instance where only those two layers where optimized in the inner loop, while the rest of the layers where shared across steps. In doing so, we found that doing so makes no difference to generalization performance, whilst increasing the training and inference speeds by at least 15 times.

