[Reviews · NeurIPS 2019]

Reviewer 1



This paper provides a learning-to-learn approach that takes advantage of the information from the test set via a transduction learning setup. Although past meta-learning literatures evaluate the few-shot generalisation performance in an implicit transduction setting, this is the first approach that trains a network explicitly to learn from unlabelled test data. It conducts extensive ablation study on the type of conditioning features for the learnt cost function, and shows SOTA performance on Mini-Imagenet and Caltech-UCSD Birds 200 benchmarks. The motivation and the algorithm of the paper is well explained. Figure 1 and Algorithm 1 are very useful to understand the proposed algorithm. What is not clear to me is how the information of the entire target set used to predict individual test images. For every image, does the loss network computes features and adapt model parameters based on the information from that image alone or from all the images of the entire target set? I guess the latter setting may give better performance since the information of other unlabelled images is useful for the image of interest and the original MAML paper also used all the test images for transductive learning. In that case, it would be important to understand how the prediction accuracy depend on the size of the target set and if it generalises to a different size from the meta-training setting. In the experiment section, the performance of SCA on Low-End MAML is not studied for the CUB dataset. It would be nice to show the results in order to show the generality of the algorithm under different settings. ------- Update after feedback: Thanks for clarification on the test batch size. It would be nice to clarify it in the final version as I imagine the performance would depend heavily on the size of test set.

Reviewer 2



Summary: This paper considers few-shot classification and seeks to make use of the unlabeled query data during few-shot classification by training on it with a meta-learned critic loss. The algorithm builds on top of MAML, and has two stages. In the first stage, the model is adapted via gradient descent on the labeled support set. In the second stage, the model is further adapted via a meta-learned critic loss that is a function of a featurization of the model parameters and the unlabeled query set. Originality: The proposed approach strikes me as quite similar to One-Shot Imitation Learning by Domain-Adaptive Meta-Learning (Yu et al. 2018). In that paper, they similarly learn a critic loss, in their case to adapt an RL agent to a human demonstration in which ground truth actions are not available. This citation should be added to the paper. More distant but still quite related is Evolved Policy Gradients (Houthhooft et al. 2018), which meta-learns a loss function for an RL agent. Quality: A few clarifying questions: What is the point of Figure 2 (shows the effect of the critic on output probabilities)? The caption is entirely descriptive of the plotted data. In no case presented does the critic actually change the top prediction! Are the network backbones the same across all comparisons in Table 2? This is very important to ensure fair comparison (see Chen et al. 2019.) In particular, SCA performs on par to LEO - are the backbones the same here? Incidentally, the idea of meta-learning a loss function is general and could be applied to other meta-learning algorithms besides MAML, I don’t see a need to restrict it to gradient-based in line 122 and other places. Clarity: The writing is clear for the most part (Sections 3 and 4 are a bit long-winded). Figure 1 is very helpful. It would be good to explain what you *expect* the critic loss to learn, for motivation. Significance: low/medium - mainly due to concerns about the validity of the results, additionally due to the overlap of the idea with previously published works. ----------------- Post-rebuttal ------------------ Thank you for clarifying the comparisons in Table 2. I feel confident now that the most important comparison (with regular MAML) is correct. I also appreciate that while it’s good to contextualize the results with respect to non-MAML based methods, this is not critical to prove the point about transduction. I’m satisfied the authors did their best in this regard. However, I would like to say that I strongly disagree with the statement, “It is fair to compare methods on the quote results on the same benchmarks.” Network backbones and training techniques improve over time, therefore it’s unreasonable to compare directly with numbers reported in older few-shot papers that built their algorithms on top of what are *now* antiquated methods. It is important that as a community we do not waste time chasing incremental improvements that are revealed to be an illusion when an older method is ported to modern times. I still feel that Figure 2 could be improved. Perhaps an analysis of how often the critic changes the predictions, or in what specific cases? “It doesn’t do nothing” is a pretty low bar for your method… Thanks for agreeing to add the requested citations. I agree that your contribution is distinct. I do think that formulating transductive meta-learning more broadly to include meta-learning besides gradient-based approaches would make the paper more impactful.

Reviewer 3



Originality: To my knowledge this is the first transductive learning MAML paper. Quality: The results are compelling. Clarity: The paper is easy to understand. Significance: The result should be interesting to anyone who works on meta learning with neural networks.

[Author Response · NeurIPS 2019]

**Response to Reviewer 6**

**Backbone clarification:** We completely agree it is critical to compare models fairly, and hope to convince you we took substantial measures to ensure there were reliable comparisons.

In Table 2, the top blocks are the published results of each particular method. Each paper chooses its own backbone, and the authors optimized the choice of model for their approach. It is fair to compare methods on the quoted results on the *same* benchmarks (which is, after all, the point of benchmarks).

On top of this we provide 3 results on 2 different backbones. The first two results are produced by the the the *same* original MAML/MAML++ backbone (Low-End), trained with either the original MAML formulation or, more reliably, the MAML++ formulation. We will clarify these are the same backbone. In addition we also provide an improved backbone for MAML++ (High-End MAML++) which utilizes a shallow but wide DenseNet with squeeze excite attention. We compare how the two MAML++ backbones perform with and without SCA to demonstrate that performance gains were due to the SCA technique. The final model can then be compared with all previous methods. The proposed combination of MAML++ and SCA and has produced the top results ever published across tasks on Mini-ImageNet and CUB to date.

We would have liked to apply SCA directly on LEO as the technique is general and can be applied to any meta-learning system. However, LEO is closed-source and we have found it very hard to reproduce even after much effort. Hence the comparisons with the MAML backbone.

**Figure 2** demonstrates there are changes to the probabilities resulting from the learnt unsupervised loss. Some top classes do change, e.g. in the rightmost chart. The effect of these changes is not in the class-labels alone but also allows better model initialization: training a critic on a pretrained initialization does not provide as much benefit. Hence we did not intend any strong conclusions beyond illustrating that the unsupervised loss has a non-trivial effect.

**Discussion on related work.** Both papers are highly relevant and we thank the reviewer for alerting us to them. We will reference them in the paper. In Yu et al. (2018), the authors learn a loss function to enable sample-efficient adaptation of an RL agent. Instead we tackle the few-shot classification task by learning a transductive model. Our model adapts a meta-learned initialization by both labelled and unlabelled examples; we do use a learnt unsupervised model, but the combination is critical. Their work is purely unsupervised in the inner loop, using RL as the outer loop, whereas ours consists of both supervised and unsupervised inner loops and a supervised outer loop.

In Evolved Policy Gradients, the author also learns a loss function to train a policy network towards an RL task. The loss function receives both states and rewards, it has an RL inner phase and an evolutionary algorithm outer loop which both also has access to rewards. We target a different problem, that is few-shot learning, using a supervised outer loop, and a transductive inner loop (composed by supervised and unsupervised inner loop phases).

**Broader formulation of transductive meta-learning:** We understand there is no particular dependence on MAML. However feedback from colleagues on earlier drafts suggested the paper clarity was improved when using a concrete case rather than the previous more general description. However we should indeed alert the reader to this generality in the conclusion.

**Response to Reviewer 3**

**Utilization of the target-set information:** We use gradients of a whole batch of target set images to update the base model since this produces better performance. The results in the paper represent experiments where we used target sets of size 75 to learn the loss function, however one could instead randomly choose the batch size of the target set to train a loss function that generalizes on a wide range of batch sizes.

**Additional diagrams:** Space constraints prevented us including further diagrams. We will prepare a longer version which will incorporate your suggestions. Code will also be available upon acceptance of this paper.

**Response to Reviewer 7**

Indeed, we are currently using the Meta-Dataset (https://arxiv.org/abs/1903.03096) in other work; however, in the context of this paper, effective runs on this were beyond our computational capacity in the time frame.

[Meta-Review · NeurIPS 2019]

The reviewers agreed that this submission makes an interesting and novel contribution to NeurIPS. I strongly encourage the authors to address the remaining comments of reviewers 3 and 6, to clarify the test batch size, improve figure 2, and add citations.